# Architecture of the human mTORC2 core complex

Edward Stuttfeld[1†], Christopher HS Aylett[2†], Stefan Imseng[1], Daniel Boehringer[2], Alain Scaiola[2], Evelyn Sauer[1], Michael N Hall[1]*, Timm Maier[1,2]*, Nenad Ban[2]*

[1]Biozentrum, University of Basel, Basel, Switzerland; [2]Institute for Molecular Biology and Biophysics, Zürich, Switzerland

**Abstract** The mammalian target of rapamycin (mTOR) is a key protein kinase controlling cellular metabolism and growth. It is part of the two structurally and functionally distinct multiprotein complexes mTORC1 and mTORC2. Dysregulation of mTOR occurs in diabetes, cancer and neurological disease. We report the architecture of human mTORC2 at intermediate resolution, revealing a conserved binding site for accessory proteins on mTOR and explaining the structural basis for the rapamycin insensitivity of the complex.
DOI: https://doi.org/10.7554/eLife.33101.001

## Introduction

The serine/threonine kinase mammalian target of rapamycin (mTOR) is the master regulator of cellular growth (*Saxton and Sabatini, 2017*). Originally identified in yeast as TOR (*Heitman et al., 1991*; *Kunz et al., 1993*), the mammalian orthologue mTOR senses a range of cellular cues, including nutrient and energy availability, and growth factor signals, to control metabolism and autophagy (*Laplante and Sabatini, 2012*; *Loewith and Hall, 2011*; *Saxton and Sabatini, 2017*). Due to its role as a central controller of cell growth, aberrant mTOR signaling is observed in major diseases (*Dazert and Hall, 2011*; *Saxton and Sabatini, 2017*). mTOR is a member of the *phosphatidylinositol-kinase-related kinase* (PIKK) family (*Keith and Schreiber, 1995*) and exerts its function as the enzymatic component of two distinct protein complexes, mTOR complex (mTORC) one and mTORC2. Together with the small protein mLST8, mTOR forms the core of both complexes (*Kim et al., 2003*; *Loewith et al., 2002*). The protein Raptor is unique to mTORC1 (*Hara et al., 2002*; *Kim et al., 2002*), whereas the proteins Rictor and SIN1 are defining subunits of mTORC2 (*Frias et al., 2006*; *Jacinto et al., 2006*; *Jacinto et al., 2004*; *Sarbassov et al., 2004*; *Yang et al., 2006*). Additionally, Protor-1/2 and Deptor can bind to mTORC2 (*Pearce et al., 2007*; *Peterson et al., 2009*). The polyketide rapamycin, in complex with the endogenous protein FKBP12, inhibits mTORC1 activity by binding the FKBP-rapamycin (FRB) domain of mTOR (*Aylett et al., 2016*; *Yang et al., 2013*). In contrast, the FKBP-rapamycin complex is unable to bind mTORC2 (*Jacinto et al., 2004*), although prolonged rapamycin treatment disrupts mTORC2 signaling by preventing mTOR incorporation into mTORC2 (*Lamming et al., 2012*; *Sarbassov et al., 2006*). mTORC2 is activated by growth factors, in particular through the insulin/PI3K signaling pathway (*Saxton and Sabatini, 2017*). The major substrates of mTORC2 are the AGC-kinase family members Akt, SGK-1 and PKC-α (*García-Martínez and Alessi, 2008*; *Ikenoue et al., 2008*; *Sarbassov et al., 2005*). Biochemical analysis has revealed that mTORCs form dimeric assemblies (*Wullschleger et al., 2005*). The architecture of mTORC1 was resolved only recently through the use of cryo-EM and X-ray crystallography (*Aylett et al., 2016*; *Yang et al., 2016*). However, although biochemical analysis and a 26 Å EM reconstruction of yeast TORC2 have suggested that TOR and Lst8 form the same dimeric core as observed in mTORC1 (*Gaubitz et al., 2015*; *Wullschleger et al., 2005*), higher resolution information on the mTORC2 assembly remains unavailable. Structural data

*For correspondence:
m.hall@unibas.ch (MNH);
timm.maier@unibas.ch (TM);
ban@mol.biol.ethz.ch (NB)

†These authors contributed equally to this work

**Competing interests:** The authors declare that no competing interests exist.

**eLife digest** To grow and multiply, a living cell must take a variety of factors into account, such as its own energy levels and the availability of nutrients. A protein called mTOR sits at the core of a signaling pathway that integrates these and other sources information. Problems with the mTOR pathway contribute to several diseases including diabetes and cancer.

The mTOR protein occurs in two distinct protein complexes, called mTORC1 and mTORC2. These complexes contain a mix of other proteins – known as accessory proteins. They also sense different cues and act upon distinct targets in the cell. Recent research reported the structure of mTORC1, which provided clues about how this complex works. Yet, much less was known about the mTORC2 complex.

Stuttfeld, Aylett et al. have now used a technique called cryo-electron microscopy to reveal the three-dimensional architecture of the human version of mTORC2. Comparing the new mTORC2 structure to the existing one for mTORC1 showed that they have many features in common but important differences too. The overall shape of both complexes is similar and each complex contains two copies of mTOR arranged in a similar way. Also, the main accessory proteins in each complex interact with almost the exact same parts of mTOR, but the accessory proteins in mTORC2 are organized differently from those of mTORC1. The different accessory proteins also have distinct shapes. These differences could help to explain why the complexes respond to different cues and recognize different targets.

These new findings provide an entry point for further studies on how mTORC2 works in cells. The next step is to get a higher resolution image of the structure of this complex to see the finer details of all the components. This may in the future help scientists to develop drugs that inhibit mTORC2 to treat cancer and other diseases.

DOI: https://doi.org/10.7554/eLife.33101.002

on mTORC2 accessory proteins are restricted to the fission yeast Sin1 CRIM domain (*Tatebe et al., 2017*) and the SIN1 PH domains (*Pan and Matsuura, 2012*).

## Results and discussion

We co-expressed the core components of human mTORC2; mTOR, mLST8, Rictor, SIN1 and Protor-1, in *Spodoptera frugiperda* cells using the MultiBac system (*Fitzgerald et al., 2006*). Expression was attained for all components, and mTORC2 was captured by affinity to a FLAG tag inserted in the amino-terminal HEAT repeats of mTOR. Reconstituted complexes were purified by size-exclusion chromatography (SEC). Yields remained low (~0.1 nmol complex per 10 L culture), as mTORC2 proved labile over the course of purification, resulting in two major species of mTOR complexes eluting from SEC; mTOR-mLST8 alone and reconstituted mTORC2 complex comprising mTOR, mLST8, Rictor, SIN1 and substoichiometric amounts of Protor-1, as confirmed by mass spectrometry (*Figure 1—figure supplement 1*). Human mTORC2 demonstrated kinase activity towards the substrate Akt-1 (*Figure 1—figure supplement 1*). mTORC2 also proved extremely labile during sample preparation for both negative staining (*Figure 1—figure supplement 2*) and cryo-electron microscopy (cryo-EM). Stabilisation of human mTORC2, by cross-linking with glutaraldehyde using a modified gradient fixation protocol, allowed us to visualise intact, rhombic complexes. We collected cryo-EM data from stabilised samples, yielding 94 953 mTOR-dimer-containing particles for single particle analysis (*Figure 1—figure supplement 3*). Due to the labile nature of the complex, we observed intact complexes only in thicker ice, which limited the achievable contrast. Single particle reconstruction of mTORC2 proceeded from both an *ab initio* model generated from negatively stained samples and from the prior structure of the *Kluyveromyces marxianus* TOR-Lst8 complex (*Baretić et al., 2016*), yielding essentially identical results. A reconstruction of non-cross-linked mTORC2 from negatively stained grids agrees well with the cryo-EM reconstructions of cross-linked mTORC2 indicating that the cross-linking procedure preserves the native structure of the complex (*Figure 1—figure supplement 2*). mTORC2 proved heterogeneous; refinement initiated from the *K.m.* TOR-Lst8 reference yielded classes encompassing free mTOR-mLST8 (30%), mTOR-mLST8 occupied on a single flank by accessory protein (52%) and a complete complex bearing accessory factors on both flanks

of the mTOR dimer (18%) (*Figure 1—figure supplement 4*). Since only a fraction of particles were of the complete complex we conclude that our mTORC2 sample dissociated prior to fixation or due to non-exhaustive cross-linking. Subsequent refinement of the complete complex with the application of $C_2$ symmetry resolved the entire complex to 7.4 Å, while a focused refinement on the accessory factor density yielded 6.2 Å resolution, based upon a gold standard FSC of 0.143 (*Figure 1—figure supplement 3*). The resolution is likely limited by the achieved contrast in the measured micrographs and the inherent flexibility of the Rictor/SIN1 domains.

The complete human mTORC2 dimer displays a general organization similar to that we previously reported for mTORC1 (*Aylett et al., 2016*) (*Figure 1*). The previous 26 Å structure of *Saccharomyces cerevisiae* TORC2 was refined without symmetry because of substantial differences in the densities resolved on either side of the central cavity (*Gaubitz et al., 2015*) (*Figure 1—figure supplement 5*), however, our complete mTORC2 complex is effectively identical (real-space two-fold rotated correlation >0.9) on either side of the symmetry axis to at least 8.9 Å (the limit of resolution in $C_1$) refined without symmetry (*Figure 1* and *Figure 1—figure supplement 3*). Both mTORC1 and mTORC2 therefore function as dimeric, $C_2$ symmetric, complexes. The dimeric arrangement of mTOR in mTORC2 is most similar to that observed for the *K.m.* TOR-Lst8 (*Baretić et al., 2016*) (*Figure 1—figure supplement 5*). The FAT domains of the (m)TOR dimer in mTORC2 and TOR-Lst8 come within 6 Å of each other, contrary to the conformation of the mTOR dimer in mTORC1 that is substantially outward-rotated. The resemblance between the free mTOR dimer and the mTOR dimer found within the mTORC2 complex may account for the increased lability of mTORC2 compared to mTORC1.

Rictor makes up most (~60%) of the mass of the mTOR accessory proteins and is predicted to have an almost entirely helical secondary structure, most of which is expected to be α-solenoidal. SIN1 consists of two domains of known structure, a CRIM domain and a PH domain, neither of which form α-solenoids. While it remains impossible to definitively trace a polypeptide chain at this resolution, we were able to identify five regions of stacked α-solenoidal repeats, of which three can be resolved to the level of individual helices and two are less well ordered, that make up the majority of the density outside of the mTOR-mLST8 dimer and account for most of the elements expected to correspond to Rictor (*Figure 2A*). The assignment of the major density features to Rictor is confirmed by a negatively stained EM reconstruction of mTORC2 lacking Protor-1, which recapitulates density for all but one partially disordered extremus of the accessory protein region (*Figure 2—figure supplement 1*). The most striking feature, which we dub the 'tower', is a well-resolved, 9/10-α-hairpin stack rising from the 'bridge' HEAT repeat of mTOR. This centrepiece is joined at one-third its length from the bridge by another well-resolved α-solenoid, encompassing 4/5-hairpins, which we refer to as the 'buttress'. The buttress links the tower to a region we term the 'body', comprising three further repeats (9–11 hairpins in total) that stack against one another, with its periphery being poorly ordered. The body is only peripherally associated with the mTOR dimer. Finally, at the top of the tower, a 'cap' region of linked density juts out toward the mTOR active site. No large regions of non-α-solenoidal density were resolved, preventing us from assigning a location to SIN1, although a small region of non-helical density at the extremity of the cap represents the most likely candidate. This location of SIN1 would corroborate previous studies implying an interaction between SIN1 and mLST8 and a role of the SIN1 CRIM domain in recruiting substrates towards the active site of mTOR within mTORC2 (*Tatebe et al., 2017*).

The principal site of interaction between Rictor and the mTOR-mLST8 dimer is the juncture of the bridge and the horn, which is occupied by the base of the tower and its contact with the buttress. This is the same site occupied by the characteristic mTORC1 subunit Raptor (*Figure 2C*). Baretic and colleagues have established that the TOR dimer interface runs along this plane, and have shown that a partner protein is not required for TOR dimer formation (*Baretić et al., 2016*). We confirm this observation for mTOR-mLST8, demonstrating that the Rictor-mTOR and Raptor-mTOR interactions can be formed only with intact mTOR dimer and are not required to stabilise the mTOR dimer. Given that the inter-subunit interfaces are completely different in the two mTOR complexes (in Raptor the cleft between the horn and bridge is occupied by the loops at the tips of three helical hairpins, whereas in mTORC2 a single helix covers the length of the cleft), there must be an as yet unknown selective advantage to a dimeric architecture.

Another less well-resolved contact is the junction of the body and the free end of the bridge amino-terminal HEAT repeats within the mTOR dimer, which does not recapitulate any contact

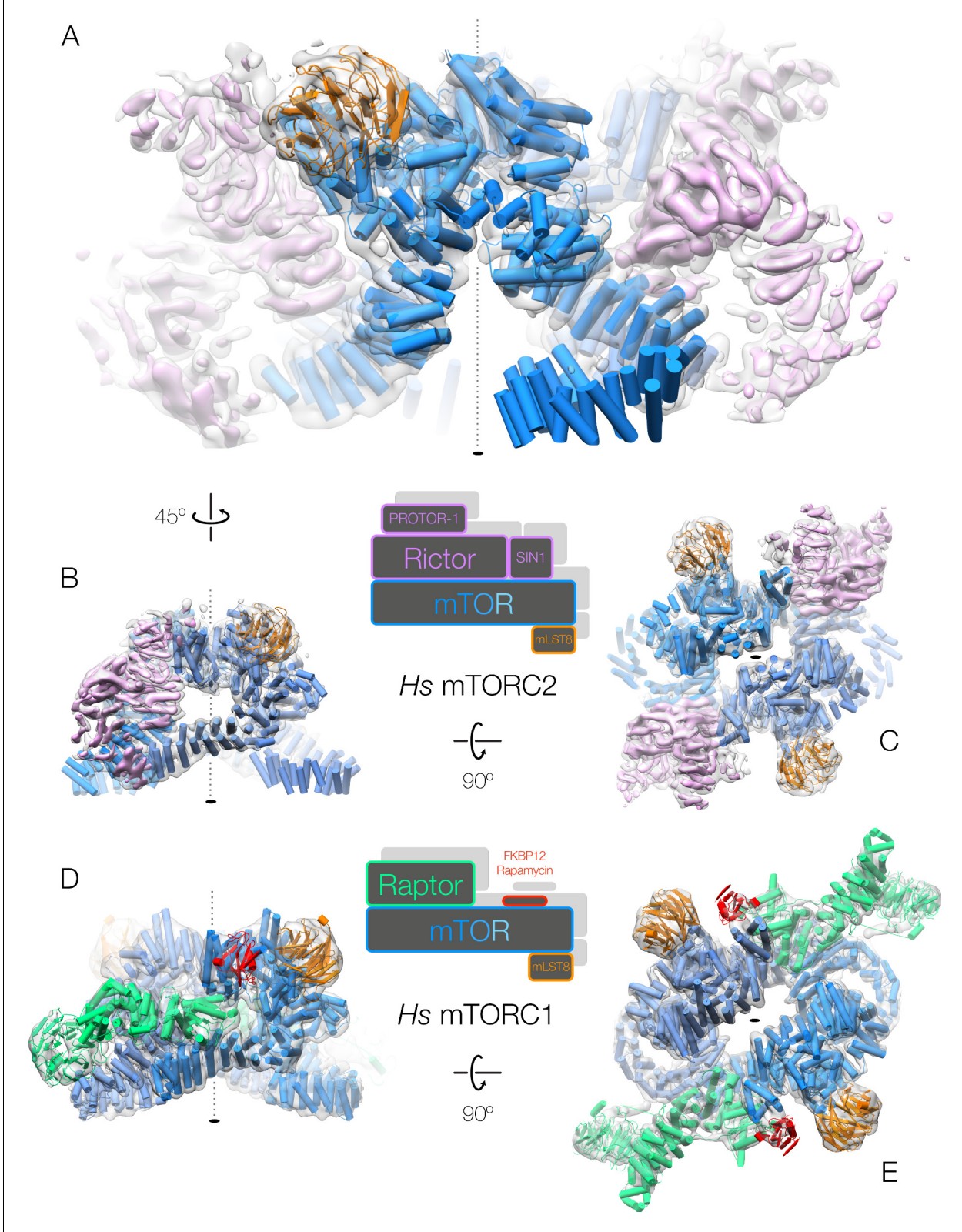

**Figure 1.** Both human mTOR complexes resolved at intermediate resolution. (**A–C**) The architecture of human mTORC2. The structure is shown rotated as indicated by the arrows between the panels. The accessory factor density from focused refinement is shown within the dimeric, $C_2$-symmetric mTORC2 density in pink. (**D–E**) The architecture of human mTORC1 (*Aylett et al., 2016*). The structure is shown rotated as indicated by the arrows

*Figure 1 continued on next page*

*Figure 1 continued*

between the panels. All complexes are shown with cryo-EM density as a grey transparent surface and the fitted structures in cartoon representation, coloured according to the primary structure schematics shown between the corresponding panels.

DOI: https://doi.org/10.7554/eLife.33101.003

The following figure supplements are available for figure 1:

**Figure supplement 1.** Purification and characterisation of mTORC2.
DOI: https://doi.org/10.7554/eLife.33101.004
**Figure supplement 2.** Analysis of negatively stained mTORC2 without applying chemical fixation.
DOI: https://doi.org/10.7554/eLife.33101.005
**Figure supplement 3.** Sample micrograph and resolution statistics for mTORC2.
DOI: https://doi.org/10.7554/eLife.33101.006
**Figure supplement 4.** Cryo-EM classification schematic.
DOI: https://doi.org/10.7554/eLife.33101.007
**Figure supplement 5.** Comparison of human and yeast TOR complexes:.
DOI: https://doi.org/10.7554/eLife.33101.008

found in mTORC1. Regions of Raptor make contact with the HEAT repeat horn in the mTORC1 structure, however, the horn does not appear to make an analogous contact with any mTORC2 accessory protein. The differences between mTORC1 and mTORC2 in the mTOR-exposed surfaces of the horn region are presumably linked to their distinct modes of regulation by post-translational modifications or binding of associated protein factors.

The final major contact identified between Rictor and the mTOR dimer is near the FRB domain of mTOR. The tower α-solenoid repeat bridges the gap between the binding site at the juncture of the mTOR dimer and the FRB domain of mTOR itself, while the cap, which is connected to the tower, straddles the surface of the FRB, deepening the active site cleft of the mTOR kinase domain. Raptor similarly deepens the active site in mTORC1, but from a position directly opposite mLST8, as opposed to Rictor, which approaches from the other side of the active site. This position of the cap masks the FKBP-rapamycin binding site of mTOR, thereby explaining the rapamycin insensitivity of mTORC2 (*Figure 2B*). This finding is consistent with what was seen previously with yeast TORC2 (*Gaubitz et al., 2015*). It is tempting to conclude that this region is responsible for substrate recruitment, given its proximity to the active site, however, further study is required to confirm this.

The architecture of human mTORC2 reveals that the two mTOR complexes share many features, including the preservation of effective $C_2$ symmetry, the common binding site for the principal accessory proteins Raptor and Rictor, the deepening of the active site cleft and the significant distance between the presumptive substrate-selection region and the active site of mTOR. While this manuscript was under revision, another publication described the architecture of *yeast* TORC2 at a resolution of 7.9 Å (*Karuppasamy et al., 2017*). The main conclusions drawn in that publication are consistent with our findings. Many important biological questions remain, in particular the reason for the requirement for $C_2$ symmetry, and the exact identity and function of the mTORC2 domains making up the cap region. A full interpretation of human mTORC2 must await future higher-resolution studies of a more stable form of the complex.

## Materials and methods

### Laboratory materials

Unless otherwise stipulated, Sf21 insect cell lines and media were provided by Expression Systems (Davies, USA) and Lonza (Basel, Switzerland), respectively. Cell lines have been tested negative for mycoplasma contamination using the MycoAlert mycoplasma detection kit from Lonza (Basel, Switzerland). Chromatography equipment was provided by GE Healthcare (Schenectady, USA), chemicals were provided by Sigma Aldrich (St. Louis, USA) and Applichem (Darmstadt, Germany), electron microscopy consumables were provided by Agar Scientific (Stansted, UK), and molecular graphics were generated using Chimera (*Pettersen et al., 2004*), Coot (*Emsley et al., 2010*) and PyMol (Schrödinger LLC, New York, USA).

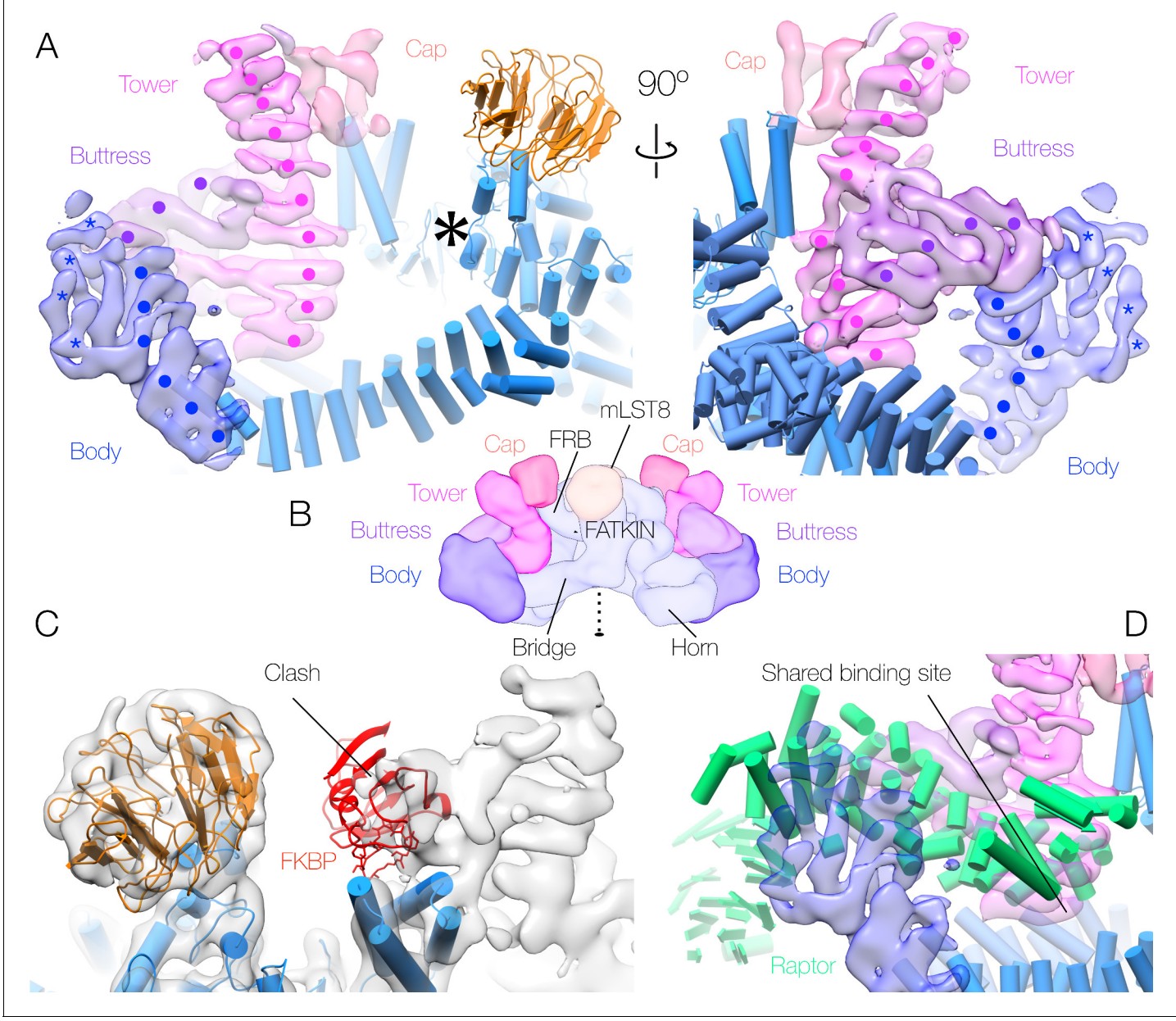

**Figure 2.** Human mTORC2 accessory factor density and binding sites. (A) Accessory factor density bound to the mTOR dimer. The regions of α-solenoid are indicated by surface colour, and the visible helical hairpins denoted by points ('•' - well-ordered regions believed to correspond to Rictor) or 5-pointed stars ('*' - poorly ordered regions that cannot be definitively assigned to Rictor or Protor-1). mTOR-mLST8 is shown in cartoon representation, and the mTOR active site is indicated by a large 6-pointed asterisk '*'. (B) Schematic overview of the 3-dimensional layout of the mTORC2 complex in the same colour scheme. Well-ordered α-solenoidal repeat density, almost all of which will correspond to Rictor, makes up the tower and buttress. The cap and body each consist of less well-ordered density; the body comprises density connecting to the buttress, and is believed to contain both regions of Rictor and peripheral density belonging to Protor-1 (see Supplement 1), whereas the cap contains density continuing from the tower, which is believed to consist of regions of Rictor and possibly SIN1. (C) Superimposition of the mTORC2 density upon the mTORC1 model, indicating the clash between FKBP and the density corresponding to the cap (black arrow). (D) Comparison between the binding site (black arrow) of Raptor within mTORC1 (cartoon representation) and the corresponding binding site for the mTORC2 accessory factors (surface representation). All panels are coloured according to the scheme in *Figure 1*.

DOI: https://doi.org/10.7554/eLife.33101.009

The following figure supplement is available for figure 2:

**Figure supplement 1.** Assignment of density to Rictor/SIN-1.
DOI: https://doi.org/10.7554/eLife.33101.010

## Cloning, expression and purification of mTORC2

Expression plasmids were prepared by cloning an internal FLAG-tag into pAB2G-mTOR after Asp258, the coding sequence of Rictor into pIDK, SIN1 into pAB1G and mLST8 into pIDC. A triple pIDC expression plasmid encoding mLST8, SIN1 and Protor-1 was synthesized de novo (Genscript, Piscataway, USA). Rictor was originally amplified from myc-Rictor corrected, which was a gift from David Sabatini (*Sarbassov et al., 2004*) (Addgene plasmid #11367). In order to clone tandem-tomato (tdTomato) labeled SIN1, the tdTomato cDNA was amplified from tdTomato-N1 and inserted into pAB1G-SIN1 using the In-Fusion HD cloning kit (Takara Bio USA, Mountain View, USA). tdTomato-N1 was kindly provided by M. Davidson and R. Tsien (Addgene plasmid #54642).

The complex of *H.s.* mTORC2 was expressed in Sf21 cells using the 'MultiBac' Baculovirus expression system (*Fitzgerald et al., 2006*) (Geneva Biotech, Geneva, Switzerland). Briefly, expression plasmids encoding FLAG-tagged mTOR, Rictor and the triple expression plasmid encoding SIN1, mLST8, and Protor-1 were fused to a 'MultiBac' expression plasmid using Cre-recombinase (New England Biolabs, Ipswich, USA) and recombined with a bacmid for baculovirus production. In order to express mTORC2-ΔProtor-1 the plasmids encoding Flag-tagged mTOR, Rictor and mLST8 were fused to a 'MultiBac' expression plasmid and recombined with a bacmid for baculovirus production. Baculovirus for the expression of tdTomato tagged SIN1 was produced separately. For the expression and purification of mTORC2-ΔProtor-1, Sf21 cells were coinfected with baculovirus encoding Flag-mTOR, Rictor and mLST8 and baculovirus encoding SIN1-tdTomato.

Sf21 cells expressing mTORC2 or mTORC2-ΔProtor-1 were harvested 72 hr post-infection and lysed in 50 mM bicine pH 8.5, 200 mM NaCl, 2 mM $MgCl_2$ lysis buffer by sonication. The lysate was cleared by centrifugation at 234 788 · g and then incubated with anti- DYKDDDDK agarose beads (Genscript, Piscataway, USA) for 2 hr at 4°C. The beads were washed 4 times in 50 mM bicine pH 8.5, 200 mM NaCl and bound protein was eluted with DYKDDDDK-peptide (0.6 mg/ml; Genscript, Piscataway, USA). The eluted protein was concentrated to 0.5 ml and applied to a tandem Superose 6 Increase 10/300 GL gel filtration column equilibrated in 10 mM bicine pH8.5, 150 mM NaCl, 0.5 mM EDTA, 2 mM TCEP. Fractions corresponding to mTORC2 were pooled, concentrated and directly used for further experiments.

## Kinase activity assay

mTORC2 kinase activity assays were conducted in 25 mM HEPES pH 7.4, 100 mM potassium acetate, 2 mM $MgCl_2$ using Akt-1 (Jena Bioscience, Jena, Germany) as a substrate. The indicated gel filtration fractions of a mTORC2 preparation were pooled and concentrated. In a 50 µl reaction volume 8.5 µg mTORC2 or mTOR/mLST8 were mixed with 100 ng inactive Akt-1 in reaction buffer and where indicated with 20 µM Torin1. The mixture was pre-incubated for 5 min on ice, and the reaction was initiated with the addition of 50 µM ATP. After 20 min at 37°C the reaction was terminated by the addition of 12.5 µl 5x SDS loading buffer. The reactions were Western Blotted using primary antibodies against phospho-Akt-Ser473 (#4060), Akt (#4685), and mTOR (#2972; Cell Signaling Technologies, Beverly, USA) at a dilution of 1:1000. A goat anti-rabbit HRP-labeled antibody (ab6721; Abcam, Cambridge, UK) was used as the secondary antibody at a dilution of 1:3000.

## Cryo-EM sample preparation and cross-linking

Given that *H.s.* mTORC2 proved highly labile, it was purified, fixed and quenched using a modification of the GraFix protocol (*Kastner et al., 2008*). Briefly, 7.5–27.5% (w/v) gradients were prepared from 150 mM NaCl, 15 mM NaBicine pH 8.0, 1 mM TCEP. The upper quarter of the less dense buffer was supplemented with 0.25% glutaraldehyde (Grade I), whereas the lower quarter of the denser buffer was supplemented with 150 mM $NH_4Cl$. Samples were subjected to ultracentrifugation for 16 hr at 100 000 · g in an SW32 rotor, thereby both fixing and quenching the sample in situ, and recovered by gradient fractionation. The recovered peak was concentrated to as low a volume as possible in 100 kDa cutoff Vivaspin concentrators (Sartorius Stedim Biotech, Aubagne, France). The buffer was then exchanged three times against 150 mM NaCl, 15 mM NaBicine pH 8.0, 1 mM TCEP in order to minimise the residual sugar in the final sample.

## Generation of the initial reference density

A fixed sample of mTORC2 was applied to a carbon coated holey carbon grid and stained with 2% (w/v) uranyl acetate. Micrographs (33) were collected using an FEI F20 electron microscope (Thermo Fisher Scientific, Waltham, MA, USA) at a magnification of 82 000 fold, an acceleration voltage of 200 kV, and a total dose of 20 e/A$^2$ at a defocus of between $-0.5$ and $-2.0$ μm. Particles (15 331) were selected semi-automatically using BOXER (*Ludtke et al., 1999*). The parameters of the contrast transfer function were then determined with CTFFIND4 (*Rohou and Grigorieff, 2015*). Particles were 2D-classified into 100 classes in two dimensions using RELION (*Scheres, 2012*) and sixteen well-defined classes were selected for initial three-dimensional reconstruction. Initial models were created using the e2initialmodel.py function in EMAN 2.1 (*Tang et al., 2007*), then filtered to 60 Å and used as an initial reference for gold-standard refinement. The resulting initial model (resolution 26 Å, gold-standard FSC) was used for further refinement.

## Negatively stained EM reconstruction of non-crosslinked mTORC2

We prepared EM samples from the peak fraction of the final Superose 6 Increase gel filtration step by negative staining. The sample was adsorbed to a freshly glow discharged thin carbon film supported on a 200 mesh copper grid and stained with 2% (w/v) uranyl acetate solution for 1 min with one additional post-staining step. The grid was imaged on a Philips CM 100 operated at 13700 fold nominal magnification at 80 kV acceleration voltage with an Olympus Veleta camera resulting in a pixel size of 5.4 Å on the specimen level. Images were acquired at a defocus between $-0.9$ and $-1.7$ μm. Particles (13879) were selected semi-automatically using BOXER (*Ludtke et al., 1999*). Contrast transfer function parameters were determined with CTFFIND4 (*Rohou and Grigorieff, 2015*). All further processing steps were done within the RELION software (*Scheres, 2012*). Initially, particles were classified into 50 classes followed by a second round of classification into 80 classes (*Figure 1—figure supplement 2*). Well-defined average images were selected, and the subset of particles which contributed to these average images was chosen for further processing. An initial model was created without applying symmetry using the 3D initial model function of RELION, which utilizes a stochastic gradient descent algorithm for de novo structure determination. The initial model was refined and used in a 3D classification into three classes applying C2 symmetry. One class (1261 particles) showed well-defined density for the core and the peripheral domains (*Figure 1—figure supplement 2*). This reconstruction (29 Å resolution) was superimposed on the cryo-EM reconstruction using UCSF CHIMERA (*Pettersen et al., 2004*).

## mTORC2-ΔProtor-1 negatively stained EM reconstruction

We prepared EM samples from affinity-purified mTORC2-ΔProtor-1 by crosslinking and negative staining. Briefly, a sample of the mTORC2-ΔProtor-1 complex was stabilised by fixation with 0.1% (v/v) glutaraldehyde for 15 min on ice. The sample was adsorbed to a thin carbon film and stained with 2% (w/v) uranyl acetate solution for 2 min with three additional post-staining steps. The sample was imaged on a Tecnai F20 electron microscope operated at 82 000 fold magnification at 200 kV acceleration voltage. Images were acquired under low dose conditions with a total dose of 20 e/Å$^2$ at $-1$ to $-2$ μm defocus on a Gatan US 4 000 CCD camera. A total of 35 micrographs were selected for further processing in Relion2 (*Scheres, 2012*) using contrast transfer function correction with CTFFIND4 (*Rohou and Grigorieff, 2015*). Single particle images were picked with Relion using a gaussian picking reference. After two-dimensional classification 13 170 single particle images were selected for refinement, during which C$_2$ symmetry was applied. The structure of the mTORC2-ΔProtor-1 complex was refined to a final resolution of 21 Å (gold-standard FSC, 0.143 criterion) (*Scheres, 2012*).

## Cryo-EM data collection

Samples of mTORC2 were applied to holey carbon copper grids (R2/2 – Quantifoil) bearing an additional fine film of carbon. Grids were blotted for two seconds and then plunged directly into a mixture of liquid ethane (33%) and propane (67%) using a vitrobot mark 4 (Thermo Fisher Scientific, Waltham, MA, USA) at 4°C and 95% humidity. Data were recorded semi-automatically using SerialEM (*Mastronarde, 2005*) on a Titan Krios transmission electron microscope (Thermo Fisher Scientific, Waltham, MA, USA) equipped with a K2 Summit direct electron detector (GATAN, San Diego,

USA) at 300 kV, 47 100 fold magnification and with an applied defocus of between −1.0 and −3.0 μm, resulting in 3 997 images of 3 838 by 3 710 pixels with a pixel size of 1.06 Å on the object scale. Each image was recorded as forty separate frames in electron counting mode, comprising a total exposure of 80 $e^-Å^{-2}$, which were subsequently aligned, summed and weighted by dose according to the method of Grant and Grigorieff (*Grant and Grigorieff, 2015*) using Motioncor2 (*Zheng et al., 2017*) to obtain the final image.

## Cryo-EM data processing and refinement

Poor quality micrographs were rejected based on the diminished maximum resolution and regularity of the Thon rings observed in the power spectra. Estimation of the contrast transfer function was carried out for each image using CTFFIND4 (*Rohou and Grigorieff, 2015*), particles were selected using BATCHBOXER (*Ludtke et al., 1999*), and refinement thereafter performed using RELION (*Scheres, 2012*). Two cycles of two-dimensional reference-free alignment of 207 920 boxed particles into 100 classes were performed initially, and particles that did not yield high-resolution class averages were excluded from further refinement. Three-dimensional classification of the remaining 94 953 mTORC2 particles into six classes was then performed with RELION (*Scheres, 2012*), separating the dataset into classes without excess density (mTOR-mLST8 - 28 774), occupied on one flank ($C_1$-mTORC2 - 48 996), and doubly occupied ($C_2$-mTORC2 - 17 183).

The $C_2$ particles were refined independently (gold-standard) using full-sized images, resulting in a complete symmetrical mTORC2 density map with an estimated resolution of 7.4 Å at a Fourier shell correlation of 0.143 (*Scheres, 2012*). Resolution was limited by the poor cryo-stability of the complex, the necessity for heavy fixation of the sample, and by conformational flexibility. Additionally, the mTOR-mLST8 core dominated the overall particle alignment, resulting in lower local resolution for the accessory proteins due to their flexibility. In order to attain the highest possible resolution of the accessory factors, the $C_1$-mTORC2 particles and the two symmetrical sides of each $C_2$-mTORC2 particle were refined in combination by subtracting the symmetry-related density and refining all occupied sides of the molecule in a single class, yielding a focused accessory factor density map with an estimated resolution of 6.2 Å at a Fourier shell correlation of 0.143 (*Scheres, 2012*).

## Data and materials availability

The cryo-EM density map representing the complete structure of mTORC2 with $C_2$ symmetry has been deposited in the EM Databank under accession code EMDB ID EMD-3927. The density corresponding to the focused refinement on the accessory proteins of mTORC2 has been deposited in the EM Databank under accession code EMDB ID EMD-3928.

# Acknowledgements

We would like to thank Michael Davidson and Roger Tsien for providing the tandem-tomato cDNA, David Sabatini for providing the myc-Rictor cDNA, the ETH Zürich scientific centre for optical and electron microscopy (ScopeM) for access to electron microscopy equipment and are indebted to P Tittmann for technical support. We would also like to thank Tim Sharpe at the Biophysics facility of the Biozentrum for SEC-MALLS analysis and the Proteomics facility of the Biozentrum for mass spectrometric protein identification.

# Additional information

## Funding

| Funder | Grant reference number | Author |
|--------|------------------------|--------|
| Schweizerischer Nationalfonds zur Förderung der Wissenschaftlichen Forschung | 51NF40_141735_NCCR | Nenad Ban |
| Schweizerischer Nationalfonds zur Förderung der Wissenschaftlichen Forschung | 310030_159492 | Michael N Hall |
| European Research Council | 609883 | Michael N Hall |

| | | |
|---|---|---|
| Sir Henry Dale Fellowship jointly funded by the Wellcome Trust and the Royal Society | 206212/Z/17/Z | Christopher HS Aylett |
| Schweizerischer Nationalfonds zur Förderung der Wissenschaftlichen Forschung | 138262 | Nenad Ban |
| Schweizerischer Nationalfonds zur Förderung der Wissenschaftlichen Forschung | 159696 | Timm Maier |
| Schweizerischer Nationalfonds zur Förderung der Wissenschaftlichen Forschung | 310030B_163478 | Nenad Ban |

The funders had no role in study design, data collection and interpretation, or the decision to submit the work for publication.

### Author contributions

Edward Stuttfeld, Conceptualization, Formal analysis, Investigation, Methodology, Writing—original draft, Writing—review and editing; Christopher HS Aylett, Conceptualization, Formal analysis, Validation, Investigation, Visualization, Methodology, Writing—original draft, Writing—review and editing; Stefan Imseng, Formal analysis, Validation, Investigation, Visualization, Writing—review and editing; Daniel Boehringer, Evelyn Sauer, Conceptualization, Formal analysis, Validation, Investigation, Visualization, Methodology, Writing—review and editing; Alain Scaiola, Data curation, Formal analysis, Validation, Investigation, Methodology, Writing—review and editing; Michael N Hall, Conceptualization, Resources, Formal analysis, Supervision, Investigation, Visualization, Project administration, Writing—review and editing; Timm Maier, Conceptualization, Resources, Formal analysis, Supervision, Funding acquisition, Investigation, Visualization, Methodology, Project administration, Writing—review and editing; Nenad Ban, Conceptualization, Resources, Supervision, Funding acquisition, Investigation, Project administration, Writing—review and editing

### Author ORCIDs

Edward Stuttfeld (iD) http://orcid.org/0000-0003-3932-9076
Christopher HS Aylett (iD) http://orcid.org/0000-0003-0662-6664
Alain Scaiola (iD) http://orcid.org/0000-0003-3233-3910
Timm Maier (iD) https://orcid.org/0000-0002-7459-1363
Nenad Ban (iD) http://orcid.org/0000-0002-9527-210X

### Decision letter and Author response

Decision letter https://doi.org/10.7554/eLife.33101.017
Author response https://doi.org/10.7554/eLife.33101.018

## Additional files

### Supplementary files

• Transparent reporting form
DOI: https://doi.org/10.7554/eLife.33101.011

### Major datasets

The following datasets were generated:

| Author(s) | Year | Dataset title | Dataset URL | Database, license, and accessibility information |
|---|---|---|---|---|
| Stuttfeld E, Aylett CHS, Imseng S, Boehringer D, Scaiola A, Sauer E, Hall MN, Maier T, Ban N | 2017 | Human mTORC2 Core Complex (C2) | http://www.ebi.ac.uk/pdbe/entry/emdb/EMD-3927 | Publicly available at the Electron Microscopy Data Bank (accession no. EMD-3927) |

| Stuttfeld E, Aylett CHS, Imseng S, Boehringer D, Scaiola A, Sauer E, Hall MN, Maier T, Ban N | 2017 | Human mTORC2 core complex accessory factors | http://www.ebi.ac.uk/pdbe/entry/emdb/EMD-3928 | Publicly available at the Electron Microscopy Data Bank (accession no. EMD-3928) |

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
