## [Decision Letter]

Thank you for submitting your article "Architecture of the Human mTORC2 Core Complex" for consideration by *eLife*. Your article has been reviewed by three peer reviewers, and the evaluation has been overseen by a Reviewing Editor and Jonathan Cooper as the Senior Editor. The following individual involved in review of your submission has agreed to reveal his identity: Kun-Liang Guan (Reviewer #1).

The reviewers have discussed the reviews with one another and the Reviewing Editor has drafted this decision to help you prepare a revised submission.

Summary:

This manuscript reports the structure of the human mTORC2 complex at intermediate resolution. The structural information reveals similarities of mTOR between mTORC1 and mTORC2 in their interactions with other subunits. Importantly, the study provides molecular basis explaining why mTORC2 is resistant to inhibition by the FKBP12-rapamycin. Therefore, this report makes an important step forward towards the molecular understanding of mTORC2 function and regulation.

Essential revisions:

1) The resolution of the mTOR2 is rather low considering the currently available instrumentation and technology. Specifically, the use of only 100K particles is unusually low for such projects. Assuming that a better resolution has not been achieved since the submission of the paper, a discussion about the technical limitations that hindered the achievement of a better resolution is necessary.

2) Can the authors provide some evidence that non cross-linked samples assume similar architecture to the presented structure? Even some noisy negative stain averages would go a long way towards convincing that cross-linking did not affect the particle architecture.

3) It is unclear how the focused refinement improved the map features – do the authors observe an improvement in features or does the improvement in value merely reflect the increase in the number of projections used?

4) The parameters reported for negative stain data collection are very similar to those used for cryo-EM. Might this be an error?

5) The authors state that "[…]our mTORC2 samples underwent partial fragmentation to yield successively less-occupied products during freezing." Do the authors mean that the complex dissociated during blotting prior to vitrification? How is this the case since the sample was cross-linked?

6) Recent findings from the cryo-EM structure of *S. cerevisiae* TORC2 (Karuppasamy et al., 2017) are also in line with some of the findings here and should therefore be mentioned in the Discussion.

7) A better schematic representation that includes the terminologies they used such as tower, bridge, horn, cap etc. should be included to better orient the reader. Is the "cap" part of rictor?

---

## [Author Response]

Essential revisions:1) The resolution of the mTOR2 is rather low considering the currently available instrumentation and technology. Specifically, the use of only 100K particles is unusually low for such projects. Assuming that a better resolution has not been achieved since the submission of the paper, a discussion about the technical limitations that hindered the achievement of a better resolution is necessary.

We assume that the thickness of ice necessary to observe intact particles combined with inherent flexibility of Rictor/SIN1 domains limits the resolution of our reconstruction. We have added a few comments in the first paragraph of the “Results and Discussion” section, stating: “Due to the labile nature of the complex we observed intact complexes only in thicker ice, which limited the achievable contrast.”, “The resolution is likely limited by the achieved contrast in the measured micrographs and the inherent flexibility of the Rictor/SIN1 domains.”

2) Can the authors provide some evidence that non cross-linked samples assume similar architecture to the presented structure? Even some noisy negative stain averages would go a long way towards convincing that cross-linking did not affect the particle architecture.

We analyzed a sample of non-cross-linked mTORC2 by negative stain EM. The class averages of the non-cross-linked mTORC2 match the reprojections of the cryo-EM structure (Figure 1—figure supplement 2). Furthermore we obtained a 3D reconstruction of the negatively stained non-cross-linked mTORC2 sample which is in agreement with the cryo-EM reconstruction (Figure 1—figure supplement 2). We added a sentence to the Results section:“A reconstruction of non-cross-linked mTORC2 from negatively stained grids agrees well with the cryo-EM reconstructions of cross-linked mTORC2 indicating that the cross-linking procedure preserves the native structure of the complex.”

3) It is unclear how the focused refinement improved the map features – do the authors observe an improvement in features or does the improvement in value merely reflect the increase in the number of projections used?

Conformational flexibility of the accessory proteins relative to the core results in lower local resolution. Indeed, the overall refinement aligned mostly the core mTOR- mLST8 and “blurred” the accessory proteins on the periphery. By focusing the refinement around the accessory proteins, ignoring the core, and combining the C1- and C2-mTORC2, we improved the local resolution and were able to improve the signal-to-noise in this area, allowing us to more readily separate the helices assigned to Rictor. We added a sentence to the Materials and methods section stating this: “Additionally, the mTOR-mLST8 core dominated the overall particle alignment, resulting in lower local resolution for the accessory proteins due to their flexibility.”

4) The parameters reported for negative stain data collection are very similar to those used for cryo-EM. Might this be an error?

This is correct as stated; a similar data collection script was used in each case.

5) The authors state that "…our mTORC2 samples underwent partial fragmentation to yield successively less-occupied products during freezing." Do the authors mean that the complex dissociated during blotting prior to vitrification? How is this the case since the sample was cross-linked?

As we have observed dissociation over all sample preparation steps, we expect the full complex to be dissociating prior and during GraFix stabilization where the sample is subjected to gradually increasing amounts of the glutaraldehyde. Therefore we would expect that the procedure resulted in cross-linking of both the dissociated complex as well as non-dissociated complex. In addition, because crosslinking is not necessarily exhaustive, we suspect additional dissociation during grid preparation. We have rewritten this section to clarify the statement: “Since only a fraction of particles were of the complete complex we conclude that our mTORC2 sample dissociated prior to fixation or due to non-exhaustive crosslinking.”

6) Recent findings from the cryo-EM structure of S. cerevisiae TORC2 (Karuppasamy et al., 2017) are also in line with some of the findings here and should therefore be mentioned in the Discussion.

We have added the reference in the Discussion in the last paragraph:“While this manuscript was under revision, another publication described the architecture of *yeast* TORC2 at a resolution of 7.9 Å (Karuppasamy et al., 2017). The main conclusions drawn in this publication are consistent with our findings.” A further extended comparison is beyond the scope and format of the current manuscript.

7) A better schematic representation that includes the terminologies they used such as tower, bridge, horn, cap etc. should be included to better orient the reader. Is the "cap" part of rictor?

We have added playdough-like schematic figure to Figure 2 as panel 2B, with the appropriate colour scheme and description in the legend.

Is the "cap" part of rictor?

We can see a connection from the tower to the cap, implying that at least a proximal part of the cap is Rictor, however the weaker density extending towards mLST8 cannot be clearly assigned. We also note that several publications place SIN1 in this region, which would as well be compatible with our data. Therefore we prefer not to make a definitive statement.